# Risk factors of hepatocellular carcinoma in type 2 diabetes patients: A two-centre study in a developing country

**Noor Atika Azit**[1], **Shahnorbanun Sahran**[2], **Leow Voon Meng**[3,4], **Manisekar Subramaniam**[4], **Suryati Mokhtar**[5], **Azmawati Mohammed Nawi**[1]*

1 Department of Community Health, Faculty of Medicine, Universiti Kebangsaan Malaysia, Cheras, Kuala Lumpur, Malaysia, 2 Faculty of Information Science and Technology, National University of Malaysia, Bangi, Selangor, Malaysia, 3 Advanced Medical and Dental Institute (AMDI), USM, Kepala Batas, Penang, Malaysia, 4 Hepato-Pancreato-Biliary Unit, Department of Surgery, Hospital Sultanah Bahiyah, Ministry of Health Malaysia, Alor Setar, Kedah, Malaysia, 5 Hepato-Pancreato-Biliary Unit, Department of Surgery, Hospital Selayang, Ministry of Health Malaysia, Batu Caves, Selangor, Malaysia

* azmawati@ppukm.ukm.edu.my

## Abstract

Type 2 diabetes mellitus (T2DM) is increasingly known as a risk factor of hepatocellular carcinoma (HCC). In this study, we determined the risk factors associated with HCC in T2DM patients. This was a matched case-control study conducted at two hepatobiliary referral centres in a developing country. Patients' sociodemographic, clinical, and biochemical characteristics between 1 January 2012 and 30 June 2018 were extracted from the electronic medical records and analysed using multivariate logistic regression analysis. A total of 212 case-control pairs were included. Significant risk factors included Chinese and Malay ethnicities that interacted with viral hepatitis (adjusted odds ratio [AOR] = 11.77, 95% confidence interval [CI]: 1.39–99.79) and (AOR = 37.94, 95% CI: 3.92–367.61) respectively, weight loss (AOR = 5.28, 95% CI: 2.29–12.19), abdominal pain/ discomfort (AOR = 6.73, 95% CI: 3.34–13.34), alcohol (AOR = 4.08, 95% CI: 1.81–9.22), fatty liver (AOR = 3.29, 95% CI: 1.40–7.76), low platelet (AOR = 4.03, 95% CI:1.90–8.55), raised alanine transaminase (AOR = 2.11, 95% CI: 1.16–3.86). and alkaline phosphatase (ALP) levels (AOR = 2.17, 95% CI: 1.17–4.00). Statins reduced the risk of HCC by 63% (AOR = 0.37, 95% CI: 0.21–0.65). The identification of these factors aids the risk stratification for HCC among T2DM patients for early detection and decision-making in patient management in the primary care setting.

## Introduction

Hepatocellular carcinoma (HCC) is one of the commonest causes of cancer-related death globally. According to global statistics, HCC-related death is showing an increasing trend. It is currently the third leading cause of cancer-related death after lung cancer and colorectal cancer [1, 2]. Furthermore, the incidence of HCC has escalated dramatically by almost 75% since 1990 [3]. It is projected to reach an even higher incidence by 2030, particularly in developing countries [4] that are undergoing rapid demographic transition [5–7].

**Data Availability Statement:** The medical record dataset retrieved and analysed in this study is not available publicly due to local regulation but it can be obtained via a written request to the Director-

General of Health Malaysia. The formal written request to the Director-General of Health Malaysia can be sent via the official DG office email: pejabatkpk@moh.gov.my.

**Funding:** This study was funded by a UKMMC Fundamental Research Grant (FF-2019-254).The funders had no role in study design, data collection and analysis, decision to publish, or preparation of the manuscript.

**Competing interests:** The authors declare no competing interest

Recently, epidemiological studies have observed a changing pattern in HCC aetiology [8]. HCC secondary to viral hepatitis has been slowly decreasing due to the widespread Hepatitis B vaccination [9]. In contrast, metabolic disease-related HCC has risen globally, as a result of an ageing population and rising prevalence of metabolic diseases. As one of the commonest metabolic diseases, diabetes mellitus (DM) has been linked with HCC. Reports have shown a 2–3-fold increase in the risk for HCC development among patients with type 2 DM (T2DM) compared with nondiabetic patients [10–12]. In view of that, there is a growing interest in elucidating the predictors among diabetic patients diagnosed with HCC to add to the known risk factors of HCC. However, most of the studies were conducted in countries with high viral hepatitis rates such as China and Taiwan [13–15] or moderate to high alcohol consumption rates such as Japan and the United Kingdom [16–19] with relatively low diabetes prevalence [20]. Furthermore, there is a lack of research in this field from developing countries that suffer from the double burden of non-communicable and communicable diseases.

Furthermore, with limited healthcare capacity, many developing countries will face a heavy toll of premature death with the double burden of disease. Effective public health interventions such as early disease detection and timely management are necessary to avert premature death [21]. To begin with, risk factors identification and stratification should be incorporated as part of early disease detection for DM patients at primary care clinics. Poor understanding of these vital factors can result in delayed detection and poor survival outcomes. In the literature, HCC patients were often diagnosed at a later stage (60.0–86.7%) [22] and many suffered from poorer overall survival than patients with other types of cancer [23]. This study aimed to determine the risk factors of HCC among T2DM patients to provide the necessary information for risk stratification of HCC in outpatient clinics. We hypothesised that the sociodemographic and clinical characteristics, as well as biochemical profiles, are associated with the risk of developing HCC among T2DM. Risk stratification based on the study findings will be helpful in early detection and prevention of misdiagnosis of HCC in the high-risk population.

## Materials and methods

### Study site and ethics statement

This study was conducted at Hospital Selayang (HS) and Hospital Sultanah Bahiyah (HSB) from 1 July 2020 to 31 December 2020. HS and HSB are among the five designated hepatobiliary referral centres in Malaysia [24]. HS, located in the state of Selangor, the country's most populous state, is the national referral centre of the hepatobiliary subspecialty. In 2015, the prevalence of diabetes in Selangor was 15.5% before rising to 18.0% in 2019. HSB is the state referral hospital for Kedah, the state that recorded the highest prevalence of diabetes in the country in 2015 (25.4%) that remained high at 24.9% in 2019 [25, 26]. This study was conducted according to ethical principles outlined in the Declaration of Helsinki and the Malaysian Good Clinical Practice Guideline. Ethical approval was obtained from the Medical Research and Ethics Committee of the Malaysian Ministry of Health (NMRR-18-3704-45037) and the National University of Malaysia Faculty of Medicine Ethics Committee (JEP-2019-356). The study was exempted from the requirement for informed consent.

### Study design and study population

This was a 1:1 matched case-control study to study the risk factors associated with HCC among T2DM patients. A total of 212 adult patients (age≥ 18 years) with newly diagnosed with HCC and with a prior diagnosis of T2DM were selected as cases based on the admission lists of the HS and HSB hepatobiliary departments from 1 January 2012 to 30 June 2018. Patients without diabetic treatment records or those with multiple cancer sites were excluded.

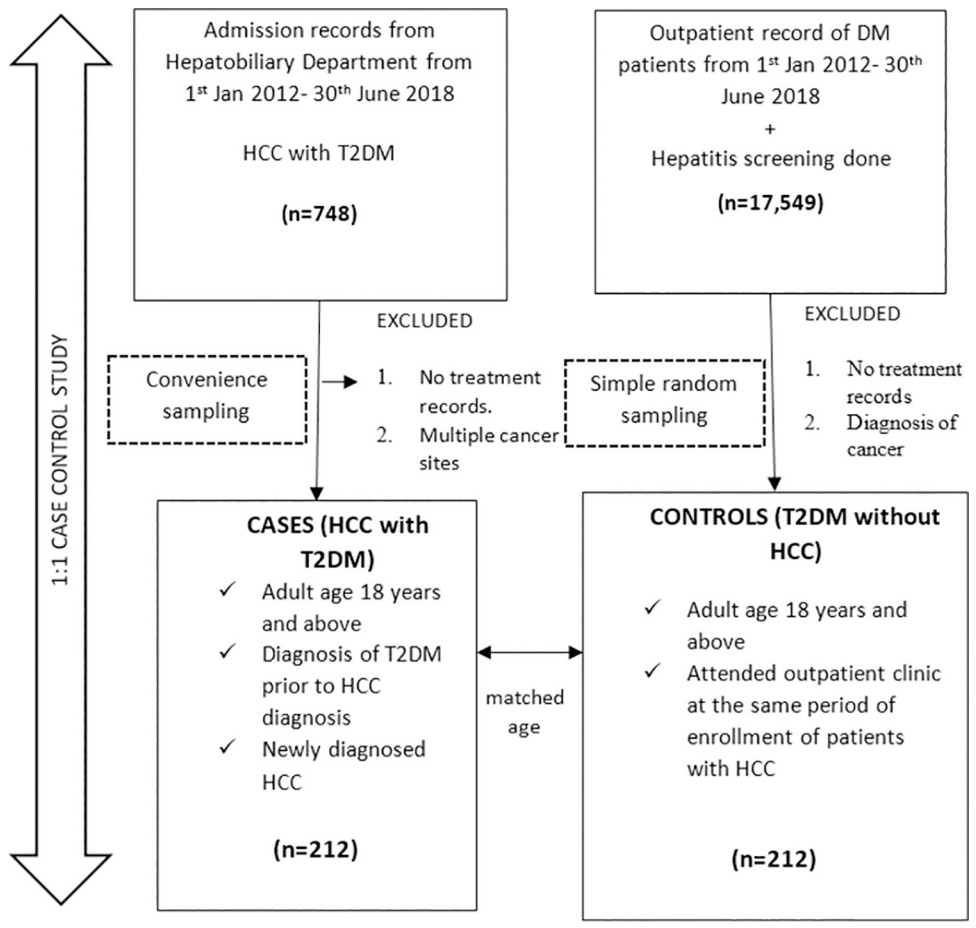

**Fig 1. The overall study design flow.**

The included cases were then matched with controls of similar age. The controls were randomly selected from the list of diabetic patients in the electronic medical records who attended the outpatient clinics between 1 January 2012 and 30 June 2018. The inclusion criteria for the controls were: adults with a known diagnosis of T2DM, attended outpatient clinics in the same year as the matched case, and had undergone hepatitis screening at the point of enrolment. Patients without diabetic treatment records and cancer diagnosis were excluded. Fig 1 illustrates the overall study flow. The sociodemographic data, clinical characteristics and investigations (biochemical parameters and imaging), and treatment data were also extracted from the electronic medical records. The sample size was calculated using Power and Sample Size Program (PS) version 3.1.2 [27]. Using a power of 80% at a 95% confidence interval (CI) (with reference to Zheng *et al.* [28]), a minimum of 78 cases was required.

## Study variables

Electronic medical records of the patients from both hospitals were reviewed to obtain the study variables. In HS, the Cerner PowerChart® Electronic Medical Record application was used for medical record management whereas HSB operated with the electronic Hospital Information System (eHIS). All the patient records can only be accessed via the workstation within the hospital's network and the user must obtain proper authority before accessing the information.

**Dependent variables.**   In this study, the outcome variable of HCC was defined as a diagnosis of HCC in the hospital record, coded as ICD-10-CM = C22.0 (International Classification of Diseases, Tenth Revision, Clinical Modification diagnostic code C22.0) and measured as a dichotomous outcome (yes, no). Certified coders from the individual hospitals performed the clinical classification coding according to the ICD-10 classification. The researcher also verified the outcome variable with the medical records of the patient. The diagnosis of HCC was made by the clinical specialist based on radiological findings from either computed tomography (CT) or magnetic resonance imaging (MRI) in accordance with the American Association for the Study of Liver Disease (AASLD) guideline [29].

**Independent variables.**   The sociodemographic data, clinical characteristics, biochemical profiles were included as independent variables in this study.

The sociodemographic characteristics included matched variables of patients: age (years), calculated based on the date of birth in the year of diagnosis (case) or year of enrolment (control); sex (male, female); and race (Chinese, Malay, and Indian).

The clinical characteristics were based on the clinical presentations documented in the medical records at diagnosis, including weight loss, lethargy, loss of appetite, abdominal pain or discomfort, and jaundice. Liver-related comorbidities were either diagnosed by the clinicians or based on radiological reports, i.e. viral hepatitis (defined as hepatitis B or hepatitis C infection diagnosed by medical practitioners), non-alcoholic fatty liver disease (fatty liver without a history of high alcohol use), cirrhosis, and portal hypertension. Other comorbidities included hypertension (reported history of hypertension or on antihypertensive medication or blood pressure $\geq$ 140/90 mmHg for two readings in the medical records) and overweight/obesity (body mass index at the last follow-up before HCC diagnosis/time of enrolment $\geq$ 23.0 kg/m$^2$). Past medical and family history of interest included the history of blood transfusion (reported transfusion of blood or blood-related products in the medical records) and family history of malignancies (recorded family history of any cancers).

In addition, the medication history was also obtained, including the usage of DM medications at the time of diagnosis/enrolment (metformin [biguanide class], glibenclamide [sulfonylureas class], gliclazide [sulfonylureas class], insulin); lipid-lowering drugs (statins); antivirals for viral hepatitis; and traditional medication. Social history such as alcohol consumption (reported intake irrespective of the amount) and smoking was also retrieved from the records. DM-related characteristics of the cases were assessed in terms of the duration of diabetes (number of years from diabetes diagnosis to the liver cancer diagnosis) and the number of years of diabetes diagnosis to the enrolment year for the controls. It was then categorised as $\geq$10 years or <10 years. The glycated haemoglobin (HbA1c) level at the last follow-up before HCC diagnosis (cases) or enrolment (controls) was obtained and categorised as $\geq$8.5% or <8.5%.

As for the biochemical profiles, the data were retrieved from the routine blood investigations conducted in the primary care setting. The investigations included full blood count (white blood cells [WBC, $\times 10^3$/μL; >11, $\leq$ 11], red blood cells [RBC, $\times 10^6$/μL; high: $\geq$4 in males, $\geq$3.5 in females; low: <4 in males, <3.5 in females], Hb [g/dL; $\geq$12, <12], platelets [$\times 10^3$/μL, <150, $\geq$150], mean platelet volume [MPV, fL; >11, $\leq$11]). Liver function test included albumin/globulin ratio (AGR, <1.1, $\geq$1.1), total bilirubin (TBil, μmol/L; $\geq$21, <21), alkaline phosphatase (ALP, IU/L; >129, $\leq$129), and alanine transaminase (ALT, IU/L; $\geq$25, <25). The coagulation profile included international normalised ratio (INR; >1.2, $\leq$1.2) while the renal profile included creatinine level, μmol/L (low: <45 in females, <59 in males; normal: 59–104 in males, 45–84 in females; high: >104 in males, >84 in females).

### Data analysis

All statistical analyses were performed using IBM Statistical Package for the Social Sciences (SPSS) version 21.

**Missing data.** As many retrospective studies commonly involve missing data, thus missing data processing was applied in this study. Variables with > 20% missing values were not included in the analysis. These missing data may reflect the fact that they were not routinely collected during clinical consultation. Thus, they might not be practical to be incorporated in the predictive model for the clinical setting. However, for the analysis purpose, multiple imputation was applied for the included variables to preserve the statistical power of the study. This method has been proven to avoid bias as compared to complete case analysis [30]. Five imputed datasets were generated using a fully conditional specification algorithm. The details on the affected variables and the comparison of data analysis before and after imputation are shown in the S1 Appendix.

**Descriptive analysis.** The descriptive characteristics of the cases and controls were presented as frequencies and percentages. Any differences between the matching variables were analysed using Pearson's chi-square test.

**Univariable and multivariable analysis.** An unconditional logistic regression analysis was performed for the matched case-controls as the data were essentially loose-matching data and might have benefited from the advantages of standard analysis[31, 32]. All variables with $p < 0.25$ were included in the multiple logistic regression analysis, with the exception of variables with < 10 observed values to avoid sparse data bias. However, all DM-related variables were included in the analysis because they were deemed as clinically important variables. The significance level was established at $p < 0.05$. The models were checked for any multicollinearity and interactions between the included variable. The model fit was examined using the Hosmer-Lemeshow goodness-of-fit test, classification table, and the receiver operating characteristic (ROC) curve.

## Results

A total of 424 patients (212 cases and 212 controls) who met the inclusion and exclusion criteria were identified and included in the study. The age range of the study population was 37 to 92 years with a mean age of 66.9 (standard deviation [SD] 9.02). There were no significant differences between the age of the cases and controls.

Table 1 presents the descriptive statistics of the sociodemographic, clinical, and biochemical profiles of the patients, as well as the univariable logistic regression results. Univariate analyses showed that being male, Chinese ethnicity, weight loss, abdominal pain or discomfort, jaundice, viral hepatitis, cirrhosis, portal hypertension, history of blood transfusion, family history of malignancy, statins, antivirals (for hepatitis treatment), history of alcohol consumption, smoking, white blood cells (WBC), platelets as well as elevated levels of total bilirubin (TBil), alkaline phosphatase (ALP), alanine transaminase (ALT), and creatinine levels were significantly associated with increased HCC risk. All variables with $p < 0.25$ were included in the multivariable analysis. Five variables (jaundice, cirrhosis, portal hypertension, family history of malignancies, antiviral treatment) were not chosen because they had <10 observed values. DM-related variables (DM medications, HbA1c, duration of DM) were included because of their clinical importance based on the medical experts' opinion.

Table 2 shows the independent predictors for developing HCC adjusted for age, sex, race, DM duration, weight loss, loss of appetite, abdominal pain/discomfort, viral hepatitis, non-alcoholic fatty liver, metformin, gliclazide, insulin, statins, blood transfusion, alcohol, smoking,

**Table 1. Univariate logistic regression analysis of risk factors for HCC among T2DM patients.**

| Risk factor | | Cases, n (%) | Controls, n (%) | Crude OR | 95%CI | p-value |
|---|---|---|---|---|---|---|
| (N = 424) | | N = 212 | N = 212 | | | |
| **Sociodemographic characteristics** | | | | | | |
| Age categories (years) | | 66.9 (9.02)[a] | 66.9 (9.02)[a] | 1.00 | 0.98–1.02 | 1.000 |
| Sex | | | | | | <0.001* |
| | Male | 165 (77.8) | 129 (60.8) | 2.26 | 1.48–3.46 | |
| | Female | 47 (22.2) | 83 (39.2) | 1.00 | | |
| Race | | | | | | <0.010* |
| | Chinese | 107 (50.5) | 56 (26.4) | 3.52 | 1.85–6.71 | |
| | Malay | 86 (40.6) | 121 (57.1) | 1.31 | 0.70–2.44 | |
| | Indian | 19 (9.0) | 35 (16.5) | 1.00 | | |
| **Clinical characteristic** | | | | | | |
| *Clinical presentation* | | | | | | |
| Loss of weight | | | | | | <0.001* |
| | Yes | 48 (22.6) | 14 (6.6) | 4.14 | 2.20–7.77 | |
| | No | 164 (77.4) | 198 (93.4) | 1.00 | | |
| Lethargy | | | | | | 0.279 |
| | Yes | 36 (17.0) | 28 (13.2) | 1.34 | 0.79–2.30 | |
| | No | 176 (83.0) | 184 (86.8) | 1.00 | | |
| Loss of appetite | | | | | | 0.037* |
| | Yes | 49 (23.1) | 32 (15.1) | 1.69 | 1.03–2.77 | |
| | No | 163 (76.9) | 180 (84.9) | 1.00 | | |
| Abdominal pain or discomfort | | | | | | <0.001* |
| | Yes | 72 (34.0) | 26 (12.3) | 3.68 | 2.23–6.06 | |
| | No | 140 (66.0) | 186 (87.7) | 1.00 | | |
| Jaundice | | | | | | 0.003* |
| | Yes | 20 (9.4) | 1 (0.5) | 21.98 | 2.92–165.33 | |
| | No | 192 (90.6) | 211 (99.5) | 1.00 | | |
| *Liver-related comorbidities* | | | | | | |
| Viral hepatitis | | | | | | |
| | Yes | 104 (49.1) | 12 (5.7) | 16.05 | 8.45–30.49 | <0.001* |
| | Hepatitis B | 65 (30.7) | 11 (5.2) | | | |
| | Hepatitis C | 39 (18.4) | 1 (0.5) | | | |
| | No | 108 (50.9) | 200 (94.3) | 1.00 | | |
| Non-alcoholic fatty liver disease | | | | | | 0.181 |
| | Yes | 30 (14.2) | 21 (9.9) | 1.50 | 0.83–1.17 | |
| | No | 182 (85.8) | 191 (90.1) | 1.00 | | |
| Cirrhosis | | | | | | <0.001* |
| | Yes | 152 (71.7) | 7 (3.3) | 74.19 | 32.99–166.85 | |
| | No | 60 (28.3) | 205 (96.7) | 1.00 | | |
| Portal hypertension | | | | | | <0.001* |
| | Yes | 49 (23.1) | 3 (1.4) | 20.94 | 6.41–68.40 | |
| | No | 163 (76.9) | 209 (98.6) | 1.00 | | |
| Ascites | | | | | | <0.001* |
| | Yes | 47 (22.2) | 5 (2.4) | 11.79 | 4.58–30.32 | |
| | No | 165 (77.8) | 207 (97.6) | 1.00 | | |
| *Other comorbidities* | | | | | | |
| Hypertension | | | | | | 0.736 |

(*Continued*)

**Table 1.** (Continued)

| Risk factor (N = 424) | | Cases, n (%) N = 212 | Controls, n (%) N = 212 | Crude OR | 95%CI | p-value |
|---|---|---|---|---|---|---|
| | Yes | 158 (74.5) | 161 (75.9) | 1.00 | | |
| | No | 54 (25.5) | 51 (24.1) | 1.08 | 0.69–1.68 | |
| Overweight/obesity | | | | | | 0.285 |
| | No | 37 (17.5) | 29 (13.7) | 1.00 | | |
| | Yes | 175 (82.5) | 183 (86.3) | 0.75 | 0.44–1.27 | |
| *Past medical and family history* | | | | | | |
| History of blood transfusion | | | | | | 0.027* |
| | Yes | 38 (17.9) | 22 (10.4) | 1.89 | 1.07–3.32 | |
| | No | 174 (82.1) | 190 (89.6) | 1.00 | | |
| Family history of malignancies | | | | | | 0.001* |
| | Yes | 23 (10.8) | 4 (1.9) | 6.32 | 2.15–18.63 | |
| | No | 189 (89.2) | 208 (98.1) | 1.00 | | |
| **Medications** | | | | | | |
| *DM medication* | | | | | | |
| Metformin | | | | | | 0.921 |
| | Yes | 130 (61.3) | 129 (60.8) | 0.98 | 0.66–1.45 | |
| | No | 82 (38.7) | 83 (39.2) | 1.00 | | |
| Gliclazide | | | | | | 0.765 |
| | Yes | 81 (38.2) | 84 (39.6) | 0.94 | 0.64–1.39 | |
| | No | 131 (61.8) | 128 (60.4) | 1.00 | | |
| Glibenclamide | | | | | | 0.212 |
| | Yes | 9 (4.2) | 15 (7.1) | 0.58 | 0.25–1.36 | |
| | No | 203 (95.8) | 197 (92.9) | 1.00 | | |
| Insulin | | | | | | 0.132 |
| | Yes | 71 (33.5) | 86 (40.6) | 0.73 | 0.50–1.10 | |
| | No | 141 (66.5) | 126 (59.4) | 1.00 | | |
| *Lipid-lowering* | | | | | | |
| Statins | | | | | | <0.001* |
| | Yes | 68 (32.1) | 144 (67.9) | 0.22 | 0.15–0.34 | |
| | No | 144 (67.9) | 68 (32.1) | 1.00 | | |
| *Others* | | | | | | |
| Antiviral (for hepatitis treatment) | | | | | | <0.001* |
| | Yes | 43 (20.3) | 2 (0.9) | 26.72 | 6.38–111.88 | |
| | No | 169 (79.7) | 210 (99.1) | 1.00 | | |
| Traditional medication | | | | | | 0.109 |
| | Yes | 32 (15.1) | 21 (9.9) | 1.62 | 0.90–2.91 | |
| | No | 180 (84.9) | 191 (90.1) | 1.00 | | |
| **Social risk factors** | | | | | | |
| History of alcohol consumption | | | | | | <0.001* |
| | Yes | 67 (31.6) | 21 (9.9) | 4.20 | 2.46–7.18 | |
| | No | 145 (68.4) | 191 (90.1) | 1.00 | | |
| Smoking | | | | | | 0.001* |
| | Yes | 104 (49.1) | 71 (33.5) | 1.91 | 1.29–2.83 | |
| | No | 108 (50.9) | 141 (66.5) | 1.00 | | |
| **DM-related characteristics** | | | | | | |
| Duration of T2DM, years | | | | | | 0.063 |

(*Continued*)

**Table 1.** (Continued)

| Risk factor (N = 424) | | Cases, n (%) N = 212 | Controls, n (%) N = 212 | Crude OR | 95%CI | p-value |
|---|---|---|---|---|---|---|
| | ≥10 | 104 (49.1) | 128 (60.4) | 1.00 | | |
| | 0–9 | 108 (50.9) | 84 (39.6) | 1.46 | 0.98–2.18 | |
| **Biochemical profile** | | | | | | |
| *DM monitoring* | | | | | | |
| HbA1c, % | | | | | | 0.656 |
| | ≥8.5 | 82 (38.7) | 78 (36.7) | 1.09 | 0.74–1.62 | |
| | <8.5 | 130 (61.3) | 134 (63.3) | 1.00 | | |
| **Full *blood* count** | | | | | | |
| WBC, ×10³/µL | | | | | | <0.001* |
| | <11 | 37 (17.5) | 76 (35.8) | 1.00 | | |
| | ≥11 | 175 (82.5) | 136 (64.2) | 2.64 | 1.68–4.16 | |
| RBC, ×10⁶/µL | | | | | | 0.106 |
| | High | 166 (78.3) | 179 (84.4) | 1.00 | | |
| | Low | 46 (21.7) | 33 (15.6) | 1.50 | 0.92–2.4 | |
| Hemoglobin, g/dL | | | | | | 0.300 |
| | ≥12 | 148 (69.8) | 286 (67.5) | 1.24 | 0.83–1.86 | |
| | <12 | 64 (30.2) | 138 (32.5) | 1.00 | | |
| Platelet, ×10³/µL | | | | | | <0.001* |
| | <150 | 83 (39.2) | 19 (9.0) | 6.54 | 3.79–11.28 | |
| | ≥150 | 129 (60.8) | 193 (91.0) | 1.00 | | |
| MPV, fL | | | | | | 0.286 |
| | >11 | 79 (37.3) | 66 (31.1) | 1.30 | 0.80–2.13 | |
| | ≤11 | 133 (62.7) | 146 (68.9) | 1.00 | | |
| *Liver function test* | | | | | | |
| AGR | | | | | | 0.052 |
| | <1.1 | 154 (72.6) | 135 (63.7) | 1.51 | 1.00–2.28 | |
| | ≥1.1 | 58 (27.4) | 77 (36.3) | 1.00 | | |
| TBil, µmol/L | | | | | | <0.001* |
| | >21 | 79 (37.3) | 30 (14.2) | 3.54 | 2.19–5.7 | |
| | ≤21 | 133 (62.7) | 182 (85.8) | 1.00 | | |
| ALP, IU/L | | | | | | <0.001* |
| | >129 | 98 (46.2) | 48 (22.6) | 2.87 | 1.93–4.47 | |
| | ≤129 | 114 (53.8) | 164 (77.4) | 1.00 | | |
| ALT, IU/L | | | | | | <0.001* |
| | ≥25 | 162 (76.4) | 87 (41.0) | 4.72 | 2.99–6.92 | |
| | <25 | 50 (23.6) | 125 (59.0) | 1.00 | | |
| *Coagulation profile* | | | | | | |
| | INR | | | | | 0.537 |
| | >1.2 | 58 (27.4) | 52 (24.5) | 1.16 | 0.73–1.84 | |
| | ≤1.2 | 154 (72.6) | 160 (75.5) | 1.00 | | |
| *Renal profile* | | | | | | |
| Creatinine level, µmol/L | | | | | | <0.001* |
| | Low | 28 (13.2) | 12 (5.7) | 4.87 | 2.29–10.38 | |
| | Normal | 134 (63.2) | 96 (45.3) | 1.90 | 1.90–4.48 | |
| | High | 50 (23.6) | 104 (49.1) | 1.00 | | |

* indicates statistically significant factors (p<0.05)

**Table 2. Independent factors associated with HCC in T2DM patients.**

| Risk factors (N = 424) | B | SE (B) | AOR[a±] | 95%CI | χ2 stat[b] | Degree of freedom | p-value |
|---|---|---|---|---|---|---|---|
| Race | | | | | 8.78 | 2 | 0.012 |
| Chinese | 1.43 | 0.51 | 4.18 | 1.53–11.4 | 7.78 | 1 | 0.005 |
| Malay | 0.74 | 0.51 | 2.09 | 0.77–5.67 | 2.08 | 1 | 0.149 |
| Indian | | | 1.00 | | | | |
| Weight loss | | | | | | | |
| Yes | 1.67 | 0.43 | 5.28 | 2.29–12.19 | 15.23 | 1 | <0.001 |
| No | | | 1.00 | | | | |
| Abdominal pain or discomfort | | | | | | | |
| Yes | 1.91 | 0.35 | 6.73 | 3.40–13.34 | 29.85 | 1 | <0.001 |
| No | | | 1.00 | | | | |
| Viral hepatitis | | | | | | | |
| Yes | -0.09 | 0.94 | 0.91 | 0.15–5.72 | 0.01 | 1 | 0.921 |
| No | | | 1.00 | | | | |
| Non-alcoholic fatty liver | | | | | | | |
| Yes | 1.19 | 0.44 | 3.29 | 1.40–7.76 | 7.41 | 1 | 0.006 |
| No | | | 1.00 | | | | |
| Statins | | | | | | | |
| Yes | -1.01 | 0.29 | 0.37 | 0.21–0.65 | 11.68 | 1 | 0.001 |
| No | | | 1.00 | | | | |
| History of alcohol consumption | | | | | | | |
| Yes | 1.41 | 0.42 | 4.08 | 1.81–9.22 | 11.48 | 1 | 0.001 |
| No | | | 1.00 | | | | |
| Platelet, ×103/µL | | | | | | | |
| <150 | 1.39 | 0.38 | 4.03 | 1.90–8.55 | 13.20 | 1 | <0.001 |
| ≥150 | | | 1.00 | | | | |
| ALP, IU/L | | | | | | | |
| >129 | 0.77 | 0.31 | 2.17 | 1.17–4.00 | 6.10 | 1 | 0.014 |
| ≤129 | | | 1.00 | | | | |
| ALT, IU/L | | | | | | | |
| ≥25 | 0.75 | 0.31 | 2.11 | 1.16–3.86 | 5.91 | 1 | 0.015 |
| <25 | | | 1.00 | | | | |
| Race × viral hepatitis | | | | | 9.86 | 2 | 0.007 |
| Chinese | 2.47 | 1.09 | 11.77 | 1.39–99.78 | 5.11 | 1 | 0.024 |
| Malay | 3.64 | 1.16 | 37.94 | 3.92–367.61 | 9.85 | 1 | 0.002 |
| Indian | | | 1.00 | | | | |

[a]AOR used multiple logistic regression with backward likelihood ratio method
[b]Wald test

traditional medication, HbA1c, WBC, RBC, platelet count, AGR, total bilirubin, ALP, ALT, INR, and creatinine.

Patients who presented with weight loss had 5.28 higher odds of developing HCC as compared to those who did not present with this symptom (95% CI: 2.29–12.19). Clinical presentation of abdominal pain or discomfort also led to 6.73 times higher odds of being diagnosed with HCC (95% CI: 3.34–13.34). Besides, patients with a history of alcohol consumption had a 4.08 higher risk of developing HCC (95% CI: 1.81–9.22) and those with NAFLD had a 3.29 times higher HCC risk (95% CI: 1.40–7.76). For the biochemical parameters, low platelet

counts ($<150 \times 10^3/\mu L$), increased ALP levels ($>129$ IU/L), and ALT ($\geq 25$ IU/L) were associated with higher odds of developing HCC (AOR = 4.03, 95% CI: 1.90–8.55; AOR = 2.17, 95% CI: 1.17–4.00; AOR = 2.11, 95% CI: 1.16–3.86, respectively). Statin use was a protective factor in which there was a 63% reduction in HCC risk compared to patients not on statins (AOR = 0.37, 95% CI: 0.21–0.65). Furthermore, there was also a significant interaction between ethnicity and viral hepatitis. Malays with viral hepatitis showed the highest odds of developing HCC while Chinese had the highest odds of developing HCC in the absence of the infection.

The model fitness was reasonably good (Hosmer Lemeshow test p-value = 0.248) and no multicollinearity problems were detected as variance inflation factor (VIF) was less than 10 for all variables. The sensitivity was 84.0%, the specificity was 85.4%, and the area under the ROC curve (AUC) was 0.920. The model accuracy was 84.2% and the Nagelkerke R Square was 64.1%. The positive and negative predictive values were 85.0% and 83.4% respectively.

## Interaction analysis

Table 3 displays the two-way interaction between the variable "ethnicity" and "viral hepatitis". In the absence of viral hepatitis infection, Chinese patients had the highest risk of HCC of 4.55 times the risk compared to Indians (95% CI: 1.60–12.96). However, in the presence of hepatitis infection, Malay patients showed the highest risk of developing HCC (AOR = 48.27, 95% CI: 3.79–615.23), followed by Chinese patients (AOR = 31.28, 95% CI: 3.02–323.84) when compared to the Indians.

When comparing within the same ethnicity, Chinese with viral hepatitis had an 11.7 times higher chance of developing HCC than Chinese without the infection. For Malays with viral hepatitis, the odds were even higher at 35.84 times compared to Malays without viral hepatitis. However, no significant difference was observed in the development of HCC between Indians with and without viral hepatitis infection. Fig 2 illustrates the interaction pattern between the different ethnicities based on the viral hepatitis status.

**Table 3. Adjusted odds ratio for the interaction between race and viral hepatitis.**

| Risk factors | AOR | 95%CI | p-value |
|---|---|---|---|
| **Viral Hepatitis = No** | | | 0.01 |
| Chinese | 4.55 | 1.60–12.96 | 0.004 |
| Malay | 2.14 | 0.76–6.08 | 0.151 |
| Indian | 1.00 | | |
| **Viral hepatitis = Yes** | | | 0.007 |
| Chinese | 31.28 | 3.02–323.84 | 0.004 |
| Malay | 48.27 | 3.79–615.42 | 0.003 |
| Indian | 1.00 | | |
| **Chinese ethnicity** | | | <0.001 |
| Viral Hepatitis = Yes | 11.70 | 3.55–38.56 | |
| Viral Hepatitis = No | 1.00 | | |
| **Malay ethnicity** | | | <0.001 |
| Viral Hepatitis = Yes | 35.84 | 8.68–148.05 | |
| Viral Hepatitis = No | 1.00 | | |
| **Indian ethnicity** | | | 0.583 |
| Viral Hepatitis = Yes | 0.45 | 0.03–7.79 | |
| Viral Hepatitis = No | 1.00 | | |

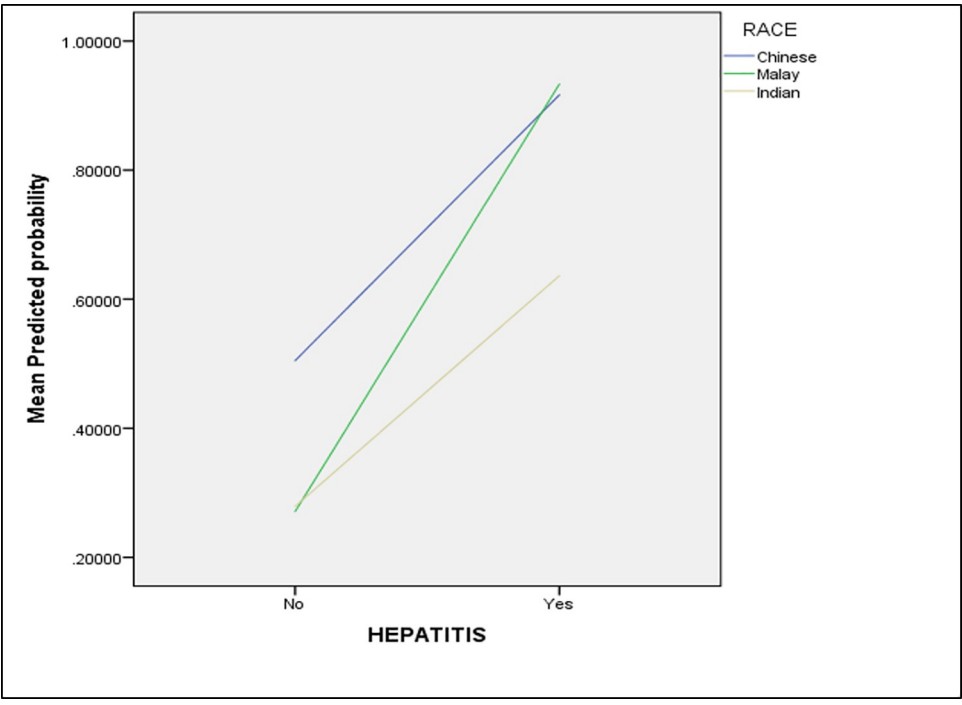

**Fig 2. Interaction plot for the predicted probability of HCC by ethnicity and viral hepatitis status.** The crossed lines showed the evidence of interaction between these factors. Chinese had a significantly ($p < 0.05$) highest risk of HCC when there is no viral hepatitis infection. However, with the presence of viral hepatitis, Malay had significantly highest odds of HCC, followed by Chinese ethnicity when compared to the Indian ethnicity ($p < 0.05$).

## Discussion

To the best of our knowledge, this is the first study to determine the risk factors of HCC among T2DM patients in a developing country with a low prevalence of chronic liver disease associated with viral hepatitis and alcohol consumption, both of which are known significant contributors of HCC development [33, 34]. This study aimed to elucidate the predisposing factors of HCC in T2DM patients to facilitate risk stratification among this group of the high-risk population.

Given the tremendous growth in DM, the study's findings are extremely relevant. DM appears to be on the rise in all world regions, with the western Pacific region having the second-highest prevalence of the disease (11.4% [95%CI; 8.3–15.9]). Malaysia has a higher age-adjusted comparative prevalence of 16.7% (95%CI; 14.9–19.2) [35]. According to the most recent Malaysian National Health Morbidity and Mortality Survey (NHMS 2019), DM prevalence was 18.3% (95%CI;17.08–19.58), and it is anticipated to rise further in the future [25]. In this multi-ethnic Asian population, DM prevalence was varied among different ethnicity. The highest prevalence was among Indian (31.4% [95%CI;25.85–37.53]), followed by Malay (21.6% [95%CI; 20.02–23.17) and Chinese (15.1% [95%CI;12.52–18.08]) [25]. Additionally, the incidence of HCC related to DM was also on the rise, especially in previously low HCC prevalence areas [36]. Diabetes/obesity accounts for 36.6% of the HCC population attributable fraction in the United States [37]. According to *Pearson-Stuttard et al.*, diabetes and a high body mass index (BMI) accounted for 24.5% of HCC cases in the general population in 2012. The researchers also demonstrated that the rising attributable risk was driven by the increased prevalence of diabetes and obesity between 1980 and 2002, implying that the incidence of HCC will continue to climb in the future [38]. Therefore, understanding the risk factors of

HCC among T2DM patients is crucial to facilitate early detection of the disease in the DM primary care setting. The miss-opportunity of early diagnosis will lead to late detection and cause a significant burden to the patients and the healthcare system.

Previously, DM was found to synergize with viral hepatitis, the main risk factor of HCC globally [10]. Interestingly, this study had elucidated that the magnitude of HCC risk differed by ethnicity among patients in a multi-ethnic Asian population. In the absence of viral hepatitis infection, Chinese ethnicity has the highest risk of HCC. This claim is supported by epidemiological evidence from prior studies which consistently revealed that Chinese ethnicity had the highest prevalence of HCC occurrence compared to Malay and Indian ethnicities.[22, 39, 40]. However, when viral hepatitis was present, Malay ethnicity was associated with the highest risk of HCC, followed by Chinese but not Indian race. The multistage carcinogenesis involving several risk factors may lead to synergy in the HCC development, contributing to a multiplicative effect on the risk estimate [41]. Moreover, viral hepatitis and DM development were influenced by genetic and ethnic variances [42, 43]. In the viral hepatitis infection, the viral-host genome integration causes chromosomal instability, mutagenesis, and activation of tumour-associated genes, resulting in uncontrolled proliferation of faulty hepatocytes that vary depending on genetic influences [44, 45]. Aside from the genetic-environmental influence, risk factors prevention, dietary composition, behavioural and biological factors differ greatly between ethnicities [46]. In this study, Malay with hepatitis has the highest odds of HCC is possible to the higher occurrence of metabolic diseases such as DM, obesity, and dyslipidaemia among Malay compared to the Chinese population [47]. In Indians, the prevalence of viral hepatitis is minimal; therefore, viral hepatitis is not a significant contributor to HCC in this group. According to a previous study, the main causes of HCC among Indians are alcohol-related and cryptogenic. Meanwhile, viral hepatitis B infection was the leading cause of HCC among Malay and Chinese people [6, 40].

Apart from the ethnic variances, in this study, the mean age of the cases (66.9 years, SD 9.02) highlighted that HCC development was more prevalent among the ageing population. This was in line with the Malaysian Cancer Registry report and previous studies of the T2DM population [11, 22]. Besides, the global epidemiological data also showed that HCC onset was associated with the socioeconomic status of a country. Patients from lower-income countries had a younger onset of HCC whereas the age of onset in middle-income countries had slowly shifted to a much older age [48].

In Malaysia, the trend of population growth indicates that we are gradually becoming an aged nation. Epidemiological transition such as this poses new challenges in the prevention and control programme against cancer such as HCC. Early diagnosis is one of the critical strategies in cancer prevention to ensure better survival. Our study findings showed that incorporating clinical presentation such as weight loss and abdominal pain/discomfort in the risk assessment can help stratify the high-risk population. Even though certain symptoms will only arise in the later stage of cancer, any suspicions of these symptoms may raise the red flag for the need for an urgent workout to prevent any delay in diagnosis. In this study, patients with abdominal pain/ discomfort and weight loss had higher odds of developing HCC. These information can be obtained during history taking at the DM outpatient clinics. Based on previous literature, abdominal pain and weight loss were among the most typical presenting symptoms of HCC patients in primary care clinics [49]. However, the abdominal pain can be non-specific with various severity from mild discomfort to severe abdominal pain depending on the underlying pathology. For instance, the patient may present with poorly localising pain arising from an enlarging lesion that stretches or irritates the viscera [50]. In contrast, acute abdominal pain may result from ruptured HCC [51]. Meanwhile, metabolic dysfunction in the advanced stage of cancer is often characterised by weight loss associated with muscle wasting and atrophy of

adipose tissue [40]. Given that these symptoms only appear at the later stage of the disease, clinicians need to integrate information about these symptoms with other risk factors to conduct a more vigilant workout to rule out malignancies and to ensure better clinical judgement in patient management.

In addition, patient's comorbidities are another vital component in the prediction of HCC. After adjusting with the other confounding factors in the multivariable analysis, we found that patients with NAFLD had higher odds of developing HCC. This association echoes the findings of many previous studies [15, 52, 53]. In the last decade, the prevalence of NAFLD has grown worldwide [54]. The complex integration of carcinogenic mechanisms in NAFLD such as chronic inflammation, lipotoxicity, and hyperinsulinaemia can lead to a higher HCC risk, especially among patients with underlying DM [55]. Statin is recommended for DM patients with NAFLD for the prevention of cardiovascular disease [56]. In the present study, the prescription of statin was found to be a protective factor against HCC, as reported in other studies [57, 58]. Several meta-analyses published in the preceding year found that statins have a consistent protective effect against HCC, with a dose-dependent relationship. This includes patients with diabetes, viral hepatitis, and cirrhosis [59–62]. Statin is a lipid-lowering drug that inhibits the 3-hydroxy-3-methylglutaryl-coenzyme A (HMG-CoA) reductase enzyme. Inhibition of HMG-CoA results in the blockage of the generation of mevalonate, a key intermediate for activation of the Ras and RhosA signalling pathways, both of which are pro-oncogenic, i.e. necessary for cell proliferation [63]. In addition, an experimental study by Relja et al. discovered that simvastatin cast an anticarcinogenic effect on the human HCC cell lines by i) inhibiting tumour cell growth, ii) inducing apoptosis iii) suppressing tumour cell cycles [64]. Therefore, statin use should be encouraged among T2DM patients.

In this study, we also found that alcohol consumption increased HCC risk by four-fold. The carcinogenic effect of alcohol on humans has long been established. In recent decades, the synergistic effect of alcohol consumption and DM status has also been widely reported [65–67]. For diabetic patients, alcohol may have directly affected the HCC development through genotoxicity or indirectly through the cirrhosis pathway. In view of this, HCC surveillance should be amplified among DM patients who consume alcohol [68].

Besides, this study discovered that low platelet count and increased liver enzyme levels (ALP and ALT) were significantly associated with the development of HCC. In previous literature, low platelet was associated with HCC development in patients with cirrhosis, especially since its high correlation with cirrhosis [69, 70]. As for the liver enzymes, ALT is primarily located in liver tissue and is responsible for the protein energy metabolism in liver cells. However, when hepatocytes are damaged, ALT will be released into the bloodstream [71]. In T2DM patients, a raised ALT level is associated with NAFLD, another risk factor of HCC [72]. In addition, several studies also reported that an increased ALP level could predict HCC recurrence after hepatectomy [73, 74]. Thus, it is used as an early predictor for HCC recurrence after liver transplantation [75]. In the bivariate analysis, lower serum creatinine was associated with HCC risk. However, the factor was not significantly associated with HCC after the adjustment of the other confounding factors. As creatinine is synthesized in the liver, low serum creatinine may be related to liver dysfunction. However, serum creatinine level was also affected by other factors such as hyper-bilirubinaemia and low nutritional status [76].

We were unable to establish a link between DM duration and HCC development in this investigation. The DM duration measured in this study may not represent the true duration due to the late identification of this chronic illness, which is one possible explanation for the finding. This assertion is supported by data from the nationwide NHMS survey, which found that the proportion of undiagnosed DM was still high in the population, at 48.9%. The subjects were unaware of the DM diagnosis before the survey [25]. Besides, the difference between

HbA1c, a biomarker for persistent hyperglycemia, is similarly not statistically significant. Previous research has yielded inconclusive results about the relationship between HbA1c levels and HCC. A large cohort research of 31,723 T2DM patients in Taiwan revealed that HbA1c fluctuation is not statistically significant between 8.5 and 17.5% but only demonstrates a statistically meaningful connection with HCC risk at >17.5% [12]. Tateishi et al. discovered in a countrywide cohort study in Japan that lower HbA1c is connected to HCC risk, in contradiction to earlier research that supports a linkage between greater HbA1c and HCC risk [13, 77–79]. The differences in the results could be attributed to unmeasured confounding factors, as HbA1c levels are altered by erythrocyte longevity, particularly in patients with chronic liver illness, chronic kidney disease, and severe metabolic abnormalities [80]. Furthermore, earlier research has shown that metformin has a preventive impact on diabetic treatments, but insulin and sulphonylureas are pro-oncogenic [81, 82]. In this study, the association was not significant. This is possible due to the combination therapy widely practiced in this country, resulting in a different direction of cancer modification effect; therefore, a pooled estimate of the cancer risk will not be significant [83].

To the best of our knowledge, this is the first study to describe the risk factors of HCC in a multiracial society with high DM prevalence and moderate burden of hepatitis infection with low alcohol consumption rate. Furthermore, the study was conducted in a developing country currently facing rapid population ageing and increasing the double burden of communicable and non-communicable diseases. Thus, the findings will provide a different perspective on the disease determinants of HCC as compared to existing studies in the literature. Besides, the existence of an interaction between ethnicity and viral hepatitis in this population has not been extensively discussed before. The similarity between the study findings and previous literature strengthens the epidemiological evidence of the risk of developing HCC among T2DM patients. Apart from that, the study finding highlighted important factors for HCC risk stratification among DM patients. The biochemical information included in this study was part of routinely collected data from T2DM patients at the DM outpatient clinics. Therefore, these criteria can be easily incorporated in the HCC risk stratification of T2DM patients in the primary care setting to aid practitioners in discovering HCC at an earlier stage and to prevent missed diagnoses. Apart from safeguarding patient care, this study also provided helpful information to improve DM and hepatitis prevention and control programmes. Another strength of this study lies in the fact this study encompassed two large hepatobiliary centres in Peninsular Malaysia that cater to patients from diverse demographic and geographic backgrounds, thus allowing a representative sample size of wide variability to ensure the reliability and generalisability of the results. Finally, the use of electronic medical record databases enabled data integration from all involved departments to ensure data precision and reduced information bias. In this country, the outpatient clinics in the hospitals also provide primary care services for DM management [84]. Future research can look to the development of risk assessment tools to facilitate HCC prediction among T2DM in the primary care setting with a prospective validation among the targeted population.

Nevertheless, there are several limitations to this research. There were missing data due to retrospective data collection. However, subjects with extensive missing data were excluded earlier on in the study to ensure the accuracy of data analysis. Besides, multiple imputation was performed to preserve the cases with minimal missing data during data analysis. Furthermore, the control group was drawn from the hospital-based outpatients that could also lead to selection bias because hospital patients might have different characteristics than the general population. For example, they might have advanced kidney disease requiring hemodialysis, thus altering their risk exposure to viral hepatitis infection [85]. In view of this, the risk estimates of the association should be interpreted carefully.

In conclusion, an in-depth understanding of the predisposing risk factors of HCC among patients with T2DM is essential in the management of HCC to ensure better survival rate. Significant parameters in this study can be used to guide HCC risk stratification in primary care facilities for better decision-making following the diagnosis of T2DM.

## Supporting information

**S1 Appendix. Missing data analysis.**
(DOCX)

## Acknowledgments

We thank the Director-General of Health Malaysia for permission to publish the research findings, the Directors of Hospital Sultanah Bahiyah and Hospital Selayang, and all information technology and medical records staff members for their assistance in this study.

## Author Contributions

**Conceptualization:** Noor Atika Azit, Azmawati Mohammed Nawi.

**Data curation:** Noor Atika Azit, Shahnorbanun Sahran, Leow Voon Meng, Manisekar Subramaniam, Suryati Mokhtar.

**Formal analysis:** Noor Atika Azit, Shahnorbanun Sahran, Azmawati Mohammed Nawi.

**Funding acquisition:** Noor Atika Azit, Azmawati Mohammed Nawi.

**Investigation:** Noor Atika Azit, Azmawati Mohammed Nawi.

**Methodology:** Noor Atika Azit, Azmawati Mohammed Nawi.

**Project administration:** Noor Atika Azit, Leow Voon Meng, Azmawati Mohammed Nawi.

**Resources:** Noor Atika Azit, Leow Voon Meng, Manisekar Subramaniam, Suryati Mokhtar, Azmawati Mohammed Nawi.

**Software:** Noor Atika Azit, Azmawati Mohammed Nawi.

**Supervision:** Shahnorbanun Sahran, Leow Voon Meng, Azmawati Mohammed Nawi.

**Validation:** Noor Atika Azit, Azmawati Mohammed Nawi.

**Visualization:** Noor Atika Azit, Azmawati Mohammed Nawi.

**Writing – original draft:** Noor Atika Azit, Shahnorbanun Sahran, Azmawati Mohammed Nawi.

**Writing – review & editing:** Noor Atika Azit, Shahnorbanun Sahran, Leow Voon Meng, Manisekar Subramaniam, Suryati Mokhtar, Azmawati Mohammed Nawi.

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
