## [Decision Letter · Decision Letter 0]

19 Oct 2021

PONE-D-21-28829Risk factors of hepatocellular carcinoma in type 2 diabetes patients: a two-centre study in a developing countryPLOS ONE

Dear Dr. Mohammed Nawi,

Thank you for submitting your manuscript to PLOS ONE. After careful consideration, we feel that it has merit but does not fully meet PLOS ONE’s publication criteria as it currently stands. Therefore, we invite you to submit a revised version of the manuscript that addresses the points raised during the review process.

Both of reviewers and I positively evaluated this manuscript, however, deeply digging the intrinsic nature, comparing the differences and similarities are necessary. We suggest the authors analyze deeply and discuss comprehensively as the comments as followings.

We look forward to receiving your revised manuscript.

Kind regards,

Yun-Wen Zheng

Academic Editor

PLOS ONE

Journal Requirements:

3. Thank you for stating the following in the Acknowledgements and Funding Section of your manuscript:

“This work was supported by the Faculty of Medicine, National University of Malaysia Medical Centre (UKMMC) (FF-2019-254).”

We note that you have provided additional information within the Acknowledgements and Funding Section. Please note that funding information should not appear in the Acknowledgments section or other areas of your manuscript. We will only publish funding information present in the Funding Statement section of the online submission form.

“This study was funded by a UKMMC Fundamental Research Grant (FF-2019-254.The funders had no role in study design, data collection and analysis, decision to publish, or preparation of the manuscript.”

Reviewers' comments:

Reviewer's Responses to Questions

**Comments to the Author**

1. Is the manuscript technically sound, and do the data support the conclusions?

Reviewer #1: Yes

Reviewer #2: Yes

2. Has the statistical analysis been performed appropriately and rigorously? 

Reviewer #1: Yes

Reviewer #2: Yes

3. Have the authors made all data underlying the findings in their manuscript fully available?

Reviewer #1: Yes

Reviewer #2: Yes

4. Is the manuscript presented in an intelligible fashion and written in standard English?

Reviewer #1: Yes

Reviewer #2: Yes

5. Review Comments to the Author

Reviewer #1: In this manuscript, the authors insist the predisposing risk factors of HCC among patients with T2DM is essential in the management of HCC to ensure better survival rate. Significant parameters in this study can be used to guide HCC risk stratification in primary care facilities for better decision making following the diagnosis of T2DM.

I have some comments.

1.There are no data on the prognosis or progression of T2DM and HCC that are the premise of this paper. In Table 1, it seems that the risk of HCC does not change depending on the duration of T2DM disease, the degree of HbA1c, and the drug used.

2.The discovery that oral statins correlate with the prognosis of HCC is very interesting. Is this happening outside of T2DM patients?

3.Table 3 shows that the risks of HCC in different races of Chinese, Malay, and Indian are quite different. It's very interesting, but these differences aren't fully considered.

4.In Figure 2, the risk of HCC due to hepatitis does not differ significantly between races. In Table 3, the risk of HCC in the presence of viral hepatitis varies considerably by race. Especially Malay's AOR seems to be quite high, what are the possible causes?

5.I would like you to discuss a little more about whether the frequency of T2DM in the area surveyed this time is higher or lower than in other areas, what kind of difference there is depending on the race, and why the analysis was focused on T2DM patients.

Reviewer #2: This manuscript describes a retrospective study of predisposing risk factor of hepatocellular carcinoma (HCC) among type 2 diabetes mellitus patient (T2DM) in a developing country.

I think this manuscript have quality required for publication in PLOSONE. However, some minor revisions are necessary for acceptance. The authors need to check the following points.

General comments.

(1) This paper discusses risk factors for HCC among T2DM patients at two hepato-biliary centers in Malaysia. And most of the results were consistent with previous studies of the HCC population with T2DM. However, it was impressive result that ethnicity was significantly involved in the difference in HCC risk for viral hepatitis, even among patients of the same yellow race. I encourage author to discuss the reason in more depth. And I would like to know if this difference is the same in patients without T2DM.

(2) Are Risk factor in Table 1 Nonalcoholic fatty disease and Risk factor in Table 2 fatty liver the same variable? If so, why does NASH show no statistical differences in univariate analysis?

(3) Several other studies cited in this article have shown the duration and severity of DM as risk factors for HCC. Please discuss why duration and HbA1c levels were not risk factors in this study, as shown in Table 1?

(4) Recently, eGFR is often used as an indicator of renal function and is considered to be superior to creatinine levels. In this paper, creatinine levels are used as the renal profile. Normal and low serum creatinine levels usually though to be better renal function than high serum creatinine levels. In Table 1, normal and low creatinine levels are shown as risk factors for HCC in univariate analysis. Would you please discuss the reason?

6. PLOS authors have the option to publish the peer review history of their article (what does this mean?). If published, this will include your full peer review and any attached files.

Reviewer #1: No

Reviewer #2: No

---

## [Author Response · Author response to Decision Letter 0]

1 Nov 2021

Dear Editor and reviewers,

Thank you for the critical assessment on our work. We are trying our best to address all the concerns, Below are the response towards the comments. We have provided a table summarizing on this in the "response to reviewers" file.

Journal Requirements:

RESPONSE : We have revise the formatting according to Plos ONE format -Page 1-37

RESPONSE : 

This study was funded by a UKMMC Fundamental Research Grant (FF-2019-254).

Financial disclosure section was deleted-Page 37

3. Thank you for stating the following in the Acknowledgements and Funding Section of your manuscript

“This work was supported by the Faculty of Medicine, National University of Malaysia Medical Centre (UKMMC) (FF-2019-254).” 

We note that you have provided additional information within the Acknowledgements and Funding Section. Please note that funding information should not appear in the Acknowledgments section or other areas of your manuscript. We will only publish funding information present in the Funding Statement section of the online submission form.

“This study was funded by a UKMMC Fundamental Research Grant (FF-2019-254.The funders had no role in study design, data collection and analysis, decision to publish, or preparation of the manuscript.”

RESPONSE : 

The statement regarding the grant had been removed from acknowledgement.

We would like to remain funding statement as previously written Page 23 

RESPONSE :

Data availability statement was in the manuscript:

 “The medical record dataset retrieved and analysed in this study is not available publicly due to local regulation but it can be obtained via a written request to the Director-General of Health Malaysia.” Page 25, line 427

Response to Reviewer #1

Reviewer #1: In this manuscript, the authors insist the predisposing risk factors of HCC among patients with T2DM is essential in the management of HCC to ensure better survival rate. Significant parameters in this study can be used to guide HCC risk stratification in primary care facilities for better decision making following the diagnosis of T2DM.

RESPONSE : 

 Thank you for your effort, time and kind review on our work. 

1.There are no data on the prognosis or progression of T2DM and HCC that are the premise of this paper. In Table 1, it seems that the risk of HCC does not change depending on the duration of T2DM disease, the degree of HbA1c, and the drug used.

RESPONSE : 

 We have added a paragraph discussing on the possible reasons of the findings.

This study was unable to elucidate the association of the DM characteristics the risk of HCC, possible due to: 

1) Duration of DM- due to the late detection of DM is still high in the population.

2) HbA1c- inconsistent findings on HBa1c association and HCC from literature. The HbA1c reading could be interfered by underlying chronic kidney and liver disease

3) DM medication- our data did not capture on the combination therapy which the pool risk may be cause the insignificant effect 

Page 23-24, line 372-390

2.The discovery that oral statins correlate with the prognosis of HCC is very interesting. Is this happening outside of T2DM patients?

RESPONSE :

 The consistent protective factors were found in other groups outside DM, based on previous literatures. We have added the statement on this with several references from previous meta-analyses 

Page 21, line 343-345

3.Table 3 shows that the risks of HCC in different races of Chinese, Malay, and Indian are quite different. It's very interesting, but these differences aren't fully considered.

RESPONSE : 

 We have added discussion on the variations. Page 20, line 291-310

4.In Figure 2, the risk of HCC due to hepatitis does not differ significantly between races.

RESPONSE : 

 Figure 2 is the interaction plot, which was not showing the significant test. As interaction plot can present a random sample error, it needs to be combined with the p-value (in table 3 ) to discard the noises (Frost, 2020). We have added caption on the figure 2 for clarity. Page 18, line 262-266

In Table 3, the risk of HCC in the presence of viral hepatitis varies considerably by race. Especially Malay's AOR seems to be quite high, what are the possible causes?

RESPONSE : 

 We also added the discussion on the elevated risk in Malay with the presence of viral hepatitis.

High AOR is possible due to multistage cancer development, contributed by multiple risk factors which result the multiplicative interaction. 

Page 20, line 291-310

5.I would like you to discuss a little more about whether the frequency of T2DM in the area surveyed this time is higher or lower than in other areas, what kind of difference there is depending on the race, and why the analysis was focused on T2DM patients.

RESPONSE : 

 Discussion was added 

Prevalence of T2DM in Malaysia was higher than global prevalence, the prevalence of T2DM in the study location (Selangor and Kedah) were among the highest in Malaysia (stated on page 5 for the study site prevalence )

Indian were the highest, followed by Malay and Chinese

Analysis was focused on T2DM patients in view of DM is in the rise, previous studies shows increasing trend in this high risk group, attributable risk to cause HCC by DM is increasing in western population, therefore need understanding on risk factors in this multi-ethnic developing country, to facilitate early detection of HCC in DM primary care Page 19, line 274-290

Response to Reviewer #2

Reviewer #2: This manuscript describes a retrospective study of predisposing risk factor of hepatocellular carcinoma (HCC) among type 2 diabetes mellitus patient (T2DM) in a developing country.

I think this manuscript have quality required for publication in PLOSONE. However, some minor revisions are necessary for acceptance. The authors need to check the following points.

RESPONSE : 

 Thank you for your effort, time and kind review on our work. 

General comments.

(1) This paper discusses risk factors for HCC among T2DM patients at two hepato-biliary centers in Malaysia. And most of the results were consistent with previous studies of the HCC population with T2DM. However, it was impressive result that ethnicity was significantly involved in the difference in HCC risk for viral hepatitis, even among patients of the same yellow race. I encourage author to discuss the reason in more depth. And I would like to know if this difference is the same in patients without T2DM. 

RESPONSE : 

Discussion was added. The effect of different ethnicities towards HCC were different in population without T2DM (Goh et al. 2015) 

Page 20, line 291-310

(2) Are Risk factor in Table 1 Nonalcoholic fatty disease and Risk factor in Table 2 fatty liver the same variable? If so, why does NASH show no statistical differences in univariate analysis? 

RESPONSE : 

It is the same variable, thank you for highlighting this. Amendment made on the fatty liver to “non-alcoholic fatty liver “, discussion on the univariate finding were added (due to existence of confounding factors, and only significant after the adjustment) Page 12, Table 1 .

Page 22, line 337

(3) Several other studies cited in this article have shown the duration and severity of DM as risk factors for HCC. Please discuss why duration and HbA1c levels were not risk factors in this study, as shown in Table 1? 

RESPONSE : 

Discussion added 

Page 23-24, line 373-391

(4) Recently, eGFR is often used as an indicator of renal function and is considered to be superior to creatinine levels. In this paper, creatinine levels are used as the renal profile. Normal and low serum creatinine levels usually though to be better renal function than high serum creatinine levels. In Table 1, normal and low creatinine levels are shown as risk factors for HCC in univariate analysis. Would you please discuss the reason? 

RESPONSE : 

Authors agree on the statement of the superiority of eGFR. However, we decided to remain the analysis as the previously presented, in view of serum creatinine may portray liver function. We have added discussion on the low creatinine associated with liver dysfunction as this may explain the bivariate analysis finding. Page 23, line 368-372

REFERENCES: 

1- Goh K-L, Razlan H, Hartono JL, Qua C-S, Yoong B-K, Koh P-S, et al. Liver cancer in Malaysia: Epidemiology and clinical presentation in a multiracial Asian population. J Dig Dis. 2015;16: 152–158. doi:10.1111/1751-2980.12223

2- Frost J. Regression Analysis; An Intuitive Guide for Using and Interpreting Linear Models. Life Course Research and Social Policies. 2020. doi:10.1007/978-3-030-36323-9_26

---

## [Decision Letter · Decision Letter 1]

15 Nov 2021

Risk factors of hepatocellular carcinoma in type 2 diabetes patients: a two-centre study in a developing country

PONE-D-21-28829R1

Dear Dr. Mohammed Nawi,

We’re pleased to inform you that your manuscript has been judged scientifically suitable for publication and will be formally accepted for publication once it meets all outstanding technical requirements.

Kind regards,

Yun-Wen Zheng

Academic Editor

PLOS ONE

Additional Editor Comments (optional):

Reviewers' comments:

Reviewer's Responses to Questions

**Comments to the Author**

1. If the authors have adequately addressed your comments raised in a previous round of review and you feel that this manuscript is now acceptable for publication, you may indicate that here to bypass the “Comments to the Author” section, enter your conflict of interest statement in the “Confidential to Editor” section, and submit your "Accept" recommendation.

Reviewer #1: All comments have been addressed

Reviewer #2: All comments have been addressed

2. Is the manuscript technically sound, and do the data support the conclusions?

Reviewer #1: Yes

Reviewer #2: (No Response)

3. Has the statistical analysis been performed appropriately and rigorously? 

Reviewer #1: Yes

Reviewer #2: (No Response)

4. Have the authors made all data underlying the findings in their manuscript fully available?

Reviewer #1: Yes

Reviewer #2: (No Response)

5. Is the manuscript presented in an intelligible fashion and written in standard English?

Reviewer #1: Yes

Reviewer #2: (No Response)

6. Review Comments to the Author

Reviewer #1: The comments are properly answered. The discussion has also been significantly added. This revised version fix makes the author's claim more understandable.

Reviewer #2: (No Response)

7. PLOS authors have the option to publish the peer review history of their article (what does this mean?). If published, this will include your full peer review and any attached files.

Reviewer #1: **Yes: **Soichiro Murata

Reviewer #2: No

---

## [Editor Report · Acceptance letter]

1 Dec 2021

PONE-D-21-28829R1 

Risk factors of hepatocellular carcinoma in type 2 diabetes patients: a two-centre study in a developing country 

Dear Dr. Mohammed Nawi:

I'm pleased to inform you that your manuscript has been deemed suitable for publication in PLOS ONE. Congratulations! Your manuscript is now with our production department. 

Kind regards, 

on behalf of

Dr. Yun-Wen Zheng 

Academic Editor

PLOS ONE